# Clinical Significance and Immune Infiltration Analyses of the Cuproptosis-Related Human Copper Proteome in Gastric Cancer

**DOI:** 10.3390/biom12101459

**Published:** 2022-10-12

**Authors:** Xiaohuan Tang, Ting Guo, Xiaolong Wu, Xuejun Gan, Yiding Wang, Fangzhou Jia, Yan Zhang, Xiaofang Xing, Xiangyu Gao, Ziyu Li

**Affiliations:** 1Key Laboratory of Carcinogenesis and Translational Research (Ministry of Education/Beijing), Department of Gastrointestinal Cancer Center Ward I, Peking University Cancer Hospital & Institute, No. 52 Fu-Cheng Road, Hai-Dian District, Beijing 100142, China; 2Biological Sample Bank, Peking University Cancer Hospital & Institute, No. 52 Fu-Cheng Road, Hai-Dian District, Beijing 100142, China

**Keywords:** gastric cancer, Cu-binding proteins, survival, immune microenvironment

## Abstract

Background: The human copper Cu proteome, also termed Cu-binding proteins (CBP), is responsible for transporting “free” Cu to the cell that is related to cuproptosis. However, their role in gastric cancer (GC) has not been reported. Methods: RNA expression data of 946 GC patients were collected. A series of machine learning and bioinformatic approaches were combined to build a CBP signature to predict survival and immune microenvironment and guide the priority treatment. Immunohistochemistry and multicolor immunofluorescence (mIF) in 1076 resection slides were used to verify immune features. Results: A CBP signature was constructed using the machine learning method from TCGA that classifies cases as CBP_low and CBP_high groups. Multivariable Cox analysis confirmed that the CBP signature was an independent prognostic factor in the training and validation cohorts. Additionally, GC patients with low CBPscores showed an increase in anti-tumor immune microenvironment, which was further verified by mIF in pathological resections following immunotherapy. Importantly, patients with low CBPscores had higher levels of TMB/MSI and responded well to immunotherapy. Conclusions: We conducted the first and comprehensive CBP analysis of GC patients and established a clinically feasible CBP signature for predicting survival and response to treatment, which will be helpful for guiding personalized medicine.

## 1. Introduction

Over the last few years, gastric cancer (GC) remains one of the most common epithelial cancers and a major source of cancer-related death [1], especially in China [2,3]. Approximately 42% of GC cases occurred in China. Furthermore, most patients are still diagnosed at an advanced or late stage and their prognosis remains poor [4]. Although the recent development of immune checkpoint inhibitors has greatly improved the treatment for GC patients, a large proportion of patients with GC cannot benefit from this schedule [5,6]. Therefore, developing effective prognostic models and methods to guide personalized medicine remains an important topic in clinical research for GC.

Previous studies have revealed that various forms of cell death were closely related to cancer progression and outcome [7]. Several novel forms of cell death, such as ferroptosis and pyroptosis, were found to play important roles in cancer eradication. For example, Badgley and coworkers found that ferroptosis inhibits pancreatic ductal adenocarcinoma growth in the mice model [8]. Feng and Yu concluded that pyroptosis inhibits the proliferation and metastasis of several types of cancer [9,10]. Recently, copper (Cu)-induced cell death was identified and termed as “Cuproptosis”, which is caused by excessive loads of “free” Cu and characterized by targeting TCA cycle proteins [11]. Furthermore, previous studies have revealed that Cu can inhibit GC growth [12] and induce anti-tumor activity [13], indicating its potential in GC treatment.

Cu is an essential element for most biological organisms and acts as a cofactor for many enzymes. An appropriate amount of cellular free Cu is important for cell biology, but too much is toxic and can result in cuproptosis. To avoid the toxicity of free Cu, organisms have specific transport systems that ‘chaperone’ the metal to targets. Recently, 54 Cu-binding proteins (CBP) are identified as the human Cu proteome, and their RNA levels are observed to be dysregulated in many cancers [14]. Among them, SLC31A1, ENOX1 and SNCA, located in the cell membrane, are responsible for cytoplasmic transport [14]. Ceruloplasmin (CP) and lysyl oxidase (LOX) are Cu-independent enzymes [15]. Anyway, the Cu-binding domain of secreted protein acidic and rich in cysteine (SPARC) is required for SPARC-mediated biologic functions [16]. Many molecules of the CBP are differentially expressed in tumor tissues and exert modulatory effects on cancer progression [17,18]. However, the expression patterns and clinical significance of most CBP in GC remain unclear. It is necessary to perform a full-scale investigation of the human Cu proteome in patients with GC.

We used STAD RNA expression data from The Cancer Genome Atlas (TCGA) to analyze the expression patterns and correlation to the survival of the human Cu proteome. Then, we constructed a CBP signature with good effects on predicting survival and immune status using machine learning methods. The median value was used as the cut-off point to enhance the randomness and accuracy. Furthermore, the CBP signature was well-validated in the ACRG cohort (*n* = 300) and data from our hospital (*n* = 198). Moreover, the CBP signature can predict response to treatment in patients with GC. Therefore, our findings revealed the expression patterns of the human Cu proteome and highlighted its potential for clinical applications to guide prognosis and treatment in patients with GC.

## 2. Materials and Methods

### 2.1. Patient Cohort and Data Collection

A total of 340 adjacent normal tissues and 946 GC samples with RNA expression data and clinicopathological information from 5 cohorts (*n* = 368, 300, 198, 35 and 45) were included in this study. Of these, the Cancer Genome Atlas (TCGA) cohort was used as the finding cohort. Additionally, the Asian Cancer Research Group (ACRG) cohort with 300 cancer samples and a cohort of 198 patients treated at the Peking University Cancer Hospital (PKUCH, Beijing, China) were used for validation. In addition, two cohorts from Sungkyunkwan University School of Medicine (SUSM, GC *n* = 45) and South San Francisco (SSF, urothelial cancer *n* = 65) treated by PD-1 blockade were used to test the predictive value of CBP signature on response to immunotherapy. Another PKUCH cohort of 35 patients who received standard perioperative chemotherapy was used to test the predictive value of CBP signature on response to chemotherapy. 

In addition, the association between the levels of CD4, CD8 and CD68 identified by tissue microarray in 1075 GC tissues from our previous study [19] and overall survival (OS) was also analyzed.

RNA expression data and clinical parameters of patients in the TCGA cohort were downloaded from the UCSC website (https://xenabrowser.net/datapages/, accessed on 1 August 2022). Correlated data of the ACRG cohort were downloaded from the Gene Expression Omnibus (GEO) database (https://www.ncbi.nlm.nih.gov/geo/, accessed on 1 August 2022). All patients provided written informed consent, and the Ethical Committee of PKUCH approved this study.

### 2.2. Establishment of the CBP-Related Signature for Prognosis

In the training set, the differentially expressed genes (DEGs) of the CBP gene set between GC and normal tissues were analyzed using the ‘limma’ R package. Then, the prognostic values of these DEGs were detected by univariate Cox regression analysis and 16 crucial genes were confirmed as prognostic factors. Then, the least absolute shrinkage and selection operator (LASSO) Cox regression model was used to screen out the most robust markers related to survival.

The generated 10 CBP genes were integrated to construct a predictive signature for the CBPscore:CBPscore=Σ (LASSO coefficient of RNAi×RNAi expression)

The best CBPscore cut-off value was determined using the median value to enhance the randomness and accuracy. The samples were classified into low- and high-CBP groups based on their median value. We then compared the OS and PFS between CBP_low and CBP_high groups to validate the prognostic predictive values using the TCGA, ACRG and PKUCH cohorts. 

### 2.3. Immunohistochemistry (IHC)

To evaluate the expression levels of CBP signature proteins, IHC results of these protein expressions in normal adjacent and GC tissues were obtained from the Human Protein Atlas (HPA, http://www.proteinatlas.org/, accessed on 1 August 2022) database and analyzed.

### 2.4. Immune Profile Analysis

Then, we analyzed tumor purity and the immune score and stromal score in cancer tissues using the ESTIMATE method [20]. Based on the global gene expression of patients, the proportion of 22-type infiltrating immune cells in each tumor was calculated using the cibersortx software (https://cibersortx.stanford.edu, accessed on 1 August 2022) [21].

### 2.5. Tumor Regression Grade (TRG) Assessment

The NCCN guidelines were used to grade tumor regression for neoadjuvant treatment in patients with AGC [22] as follows: Grade 0, complete regression with no residual tumor cells; Grade 1, near-complete response with single cells or rare small groups of cancer cells; Grade 2, partial tumor regression, with residual cancer cells with evident tumor regression, but more than single cells or rare small groups of cancer cells; Grade 3, extensive residual cancer with no evident tumor regression. Patients with TRG 0–1 were defined as responders, while non-responders with TRG 3.

### 2.6. Multiplex Immunofluorescence (mIF)

Multiplex immunofluorescence staining was conducted using the Akoya OPAL Polaris 4-Color Automation IHC kit (NEL871001KT). FFPE tissue slides were first deparaffinized in a BOND RX system (Leica Biosystems) and then incubated sequentially with primary antibodies targeting CD4 (1:100: Abcam, ab133616), CD8 (1:200, Abcam, ab178089), CD163 (1:500, Abcam, ab182422), CD68 (1:1000, Abcam, ab213363), PD-1 (1:200, CST, D4W2J, 86163S), PD-L1 (1:400, CST, E1L3N, 13684S) and CD20 (1:1, Dako, L26, IR604). After that, these slides were incubated with secondary antibodies and corresponding reactive Opal fluorophores. DAPI was used to stain nuclei acids. Tissue slides that have been bound with primary and secondary antibodies but not fluorophores were then included as negative controls to assess autofluorescence. A Vectra Polaris Quantitative Pathology Imaging System (Akoya Biosciences) was used to scan multiplex stained slides at 20-nanometer wavelength intervals from 440 nm to 780 nm. Then, the complete image for each slide was created by superimposing all scans. Multilayer images were imported to inForm v.2.4.8 (Akoya Biosciences) for quantitative image analysis. Tumor tissues and stromal tissues were differentiated by Pan-CK staining. The quantities of various cell populations were expressed as the number of stained cells per square millimeter and as the percentage of positively stained cells in all nucleated cells.

### 2.7. Statistical Analysis

Statistical analyses and figure plots were performed using the GraphPad Prism 8.0 (GraphPad Software Inc., San Diego, CA, USA) software, R software (Version 4.1.0, http://www.r-project.org, accessed on 1 August 2022) and SPSS Statistics 25 (IBM Corp., Armonk, NY, USA). RNA expression data were normalized using a Z-score method. The R ‘limma’ package was used to identify DEGs. The relationship between CBP signature and other categorical variables was analyzed using Chi-square tests. The univariate and multivariate COX analyses were performed using the SPSS software. All statistical tests were two-sided, and statistical significance was set at * *p* < 0.05.

## 3. Results

### 3.1. Construction of the CBP-Related Prognostic Signature in GC

The process of identifying the CBP-related signature was displayed in the flow chart (Appendix A). Based on the identified human Cu proteome, also termed CBP [14], we collected RNA expression profiles and analyzed their expression levels in GC and normal tissues. A total of 51 genes from the human Cu proteome were identified in the training cohort downloaded from the TCGA database (Figure 1A). Then, the differentially expressed genes (DEGs) between the normal and cancerous gastric tissues were identified using the R ‘limma’ package and 31 DEGs were screened out from the 51 CBP (Figure 1B). Meanwhile, we performed univariate Cox regression analysis and found that 16 genes of the 31 DEGs were associated with GC prognosis (Figure 1C). The collinearity among the 16 identified DEGs from the human Cu proteome was detected and identified, indicating intermolecular influences in predicting survival (Figure 1D). Subsequently, the LASSO Cox regression analysis was carried out to identify the most robust markers for the survival of GC patients (Figure 1E). Finally, an ensemble of 10 genes (LOX, CP, F5, SPARC, LOXL3, ALB, AFP, ENOX1, SLC31A1 and SNCA) had individual non-zero coefficients (Figure 1F), which were integrated to establish a CBP-related predictive signature (CBPscore).

### 3.2. Verification of the Expression of Ten CBP Signature Genes between GC and Adjacent Normal Tissues

To evaluate the expression levels of the 10 CBP genes (LOX, CP, F5, SPARC, LOXL3, ALB, AFP, ENOX1, SLC31A1 and SNCA) between the adjacent normal and GC tissues, we analyzed their mRNA expression levels in 15 normal and 35 GC tissues from our previous study [23]. The results showed that seven of the ten CBP genes were differentially expressed in normal and cancerous gastric tissues, and six genes were upregulated in GC tissues (Figure 2A). Moreover, the calculated CBPscores were significantly higher in GC tissues (Figure 2B). Then, immunohistochemical staining images from the HPA database (https://www.proteinatlas.org/) were downloaded to further confirm the expressed differences in protein expression levels. SPARC, ALB, CP, ENOX1 and AFP were highly expressed in the GC tissues compared to the adjacent normal tissues, while SLC31A1 was lower (Figure 2C). However, the expression levels of SNCA and LOXL3 were not different in GC and adjacent normal gastric tissues, and LOX and F5 were not available. These results confirmed that the ten CBP signature genes are associated with GC progression.

### 3.3. The Prognostic Value of the CBP Signature

To investigate the prognostic value of the CBP signature, we first analyzed the relationship between CBPscores and survival status in the TCGA cohort. As depicted in Figure 3A,B, a higher score of the CBP signature predicted poor survival. The median value was used as the cut-off point to enhance the randomness and accuracy. The Kaplan–Meier curve analyses revealed that patients with higher CBPscores had poor OS and PFS (Figure 3C,D). Furthermore, a time-dependent ROC analysis was performed to evaluate the predictive ability of the CBP signature for GC patients’ survival. The area under the curve (AUC) at 1-, 3- and 5-year OS were 0.64, 0.66 and 0.75, respectively (Figure 3E). Furthermore, PCA analysis was performed to descript the different distributions of CBP signature genes between low- and high-CBP groups. The results showed that patients in the two groups had different distribution patterns (Figure 3F).

### 3.4. Verification of the CBP Prognostic Signature

To further verify the prognostic value of our CBP signature, we analyzed the CBPscores in two independent cohorts, including the ACRG cohort with 300 cases and the PKUCH with 198 GC cases. Consistent with the training cohort, the cut-off value was also determined by the median CBPscore in the validation cohorts. Consistent results were identified in the ACRG cohort, a higher CBP signature score was detected in the GC tissues (Figure 4A). The Kaplan–Meier curves verified that the OS and PFS of GC patients with lower CBPscores were longer than cases with higher CBPscores (Figure 4B,C). The AUC of time-dependent ROC at 1-, 3- and 5-year OS were 0.67, 0.65 and 0.64, respectively (Figure 4D). However, PCA results did not find significantly a different distribution of patients in the two groups from the ACRG cohort (Figure 4E). The results in 198 patients from PKUCH also confirmed these findings. GC tissues had a higher CBPscore (Figure 4F), and patients with higher CBPscores had a shorter OS and PFS (Figure 4G,H). The AUC of time-dependent ROC at 1-, 3- and 5-year OS were 0.56, 0.57 and 0.57, respectively (Figure 4I). Furthermore, PCA results showed different distributions of patients in the two groups from the PKUCH cohort (Figure 4J). These results confirmed the prognostic value of the CBP signature in GC patients. 

### 3.5. Correlation Analyses between the CBP Signature and Clinical Information

We performed correlation analyses to investigate the relationship between the CBP signature and clinicopathological characteristics of GC patients. As shown in the heatmap (Figure 5A–C) and scatter diagrams (Figure 5D–J), CBP signature was significantly associated with the survival status in TCGA (*p* < 0.0001), ACRG (*p* < 0.0001) and PKUCH (*p* < 0.05); T stage in TCGA (*p* < 0.001) and ACRG (*p* < 0.001) cohorts; AJCC stage in ACRG (*p* < 0.001) and PKUCH (*p* = 0.081) cohorts. Then, we performed the univariate and multivariate Cox regression analyses to confirm the prognostic prediction of CBP signature, and the results showed that the CBP signature represents an independent risk factor in TCGA (*p* < 0.001, HR: 3.064, 95%CI: 1.945–4.826), ACTG (*p* = 0.002, HR: 1.956, 95%CI: 1.267–3.019) and PKUCH (*p* = 0.043, HR: 1.745, 95%CI: 1.398–2.179) cohorts for OS (Figure 5K–M). These results indicate that the CBP signature is an independent risk indicator for patients with GC.

### 3.6. Correlation Analyses between the CBP Signature and Immune Infiltration

To further investigate the role of CBP signature in GC, we first analyzed the subcellular localizations of CBP signature genes. According to the previous studies [11,14], nine of the ten CBP signature genes are mainly located in extracellular space, SLC31A1 and ENOX1 are located in the cellular membrane and SNCA is located in the extracellular space, cellular membrane, nucleus and cytoplasm (Figure 6A). Their subcellular localizations indicated that the CBP signature genes were associated with the tumor microenvironment. Then, we analyzed tumor purity and the presence of infiltrating immune and stromal cells in cancer tissues using the ESTIMATE method [20]. The results showed that increasing CBPscores indicated elevated immune scores, stromal scores and ESTIMATE scores and reduced tumor purity (Figure 6B).

To further understand the CBP-related immune landscape, the relationship between CBPscores and immune infiltration was analyzed. The proportions of 22-type immune infiltrated cells were calculated using a CIBERSORT algorithm [24] in the three cohorts (Figure 6C–E). The results showed that high CBPscores are companied with pro-tumor immune infiltrations including M2 macrophages and resting mast cells, and samples with low CBPscores had more anti-tumor immune infiltrations (CD8 T cells, plasma cells, memory activated CD4 T cells and follicular helper T cells) (Figure 6F–H). These results indicated that the CBP signature is associated with tumor immune infiltration.

### 3.7. Association of CBP Signature, TMB/MSI and Response to Treatment

Recently, immunotherapy based on inhibitors to immune checkpoints has been well-developed and greatly improved the prognosis of patients with GC. Recent studies revealed that tumor mutation burden (TMB) and microsatellite instability (MSI) are important factors related to response to immunotherapy in cancer [25,26]. Then, we investigated the association between CBPscores, TMB and MSI in TCGA and ACRG cohorts. The results showed that CBPscores were negatively associated with TMB and MSI (Figure 7A,B) in the TCGA cohort. Consistent with the above results, the MSI subtype in ACRG classification also had low CBPscores compared to MSS/EMT, MSS/TP53- and MSS/TP53+ types (Figure 7C). Collective, patients with low CBPscores had a higher level of TMB/MSI, which indicated they responded well to immunotherapy. Then, a GC cohort with 45 patients treated with PD-1 blockade was used to investigate the relationship between the CBP signature and therapeutic response. The results showed that responders to PD-1 blockade had lower CBPscores (Figure 7D). Due to the lack of RNA data in GC cases with immunotherapy, we evaluate this relationship in a urothelial cancer cohort (*n* = 65), while patients who received chemotherapy were eliminated. The results also showed that patients in the low CBP group had a good response to PD-1 blockade monotherapy (Figure 7E). In addition, responders to neoadjuvant chemotherapy of a PKUCH cohort (*n* = 35) had higher CBPscores compared to non-responders (Figure 7F). These results indicated that CBP signature can be used to guide personalized medicine in patients with GC.

Additionally, to further verify the immune features in patients with low CBPscores who are sensitive to immunotherapy, we performed mIF detection on postoperative pathological sections from four responders (TRG 1) and four non-responders (TRG 3) who received PD-1 blockade in our center (Figure 8A–C). Consistent with the results of immune infiltration analyses, responders (CBP_low) had a higher rate of anti-tumor immune subsets, such as elevated M1/M2 macrophage ratio (Figure 8D), increased CD8+ T cells (Figure 8E) and CD4+ T cells (Figure 8F). In addition, we analyzed the association between OS and levels of CD4, CD8 and CD68 in 1076 GC patients. The results showed that patients with a high level of these markers had a longer OS (Figure 8G–I). All these results indicated that patients with lower CBPscores had more anti-tumor cells in the TME that can be activated to exert anti-tumor effects and responded well to immunotherapy.

### 3.8. Establishment and Validation of a Nomogram Signature

Since independent prognostic analyses found that the AJCC pathological stage, age and CBP signature were independent prognostic indicators for patients with GC, we established a nomogram to evaluate the probability of 1-, 3- and 5-year OS (Figure 9A). The calibration chart revealed that the predicted OS probability by the nomogram approximated the actual probability well, and the 3-year OS prediction best agreed with the actual OS (Figure 9B–D). The Kaplan–Meier curve analysis revealed that patients with higher nomogram scores had reduced OS (Figure 9E). Furthermore, the AUC at 1-, 3- and 5-year OS were 0.68, 0.69 and 0.80, respectively (Figure 9F), better than signal AJCC pathological stage, age and CBPscores.

## 4. Discussion

Although great efforts have been made in developing comprehensive therapeutic strategies, the prognosis of patients with GC remains poor with a 5-year survival rate of <15% [27]. Further exploration of the underlying mechanisms and prognostic factors will contribute to the classification of patients and screen a better treatment strategy for GC patients. Recently, cuproptosis was revealed to play an important role in cancer development and progression [11]. Furthermore, several recent studies confirmed that signatures of cuproptosis are associated with the prognosis of patients with cancer [28,29]. However, the role of cuproptosis in GC remains unclear. These things considered, CBP is the transport system of free active Cu in cells. Then, we used CBP genes, the human Cu proteome, as the training gene set and firstly analyzed the expression patterns and correlation to the survival of GC patients. A significant correlation between CBP and GC patients’ survival was observed.

Then, we developed a CBP signature by screening the most effective prognostic genes of the human Cu proteome via LASSO and Cox regression analyses. A signature with ten CBP genes was constructed and nine of them were highly expressed and SLC31A1 was lowly expressed in GC tissues compared to the normal adjacent tissues. Among them, SLC31A1 is responsible for transporting the free Cu into intracellular space and accelerating the cuproptosis-induced cell death [11]. High levels of SLC31A1 inhibit tumor progression and SLC31A1 blockade promotes tumor development [18,30]. In addition, SPARC, LOXL3, SNCA, LOX and AFP are highly expressed and related to poor survival in GC [31,32,33,34,35]. In the established CBP signature, SLC31A1 had the only negative parameter. Therefore, patients with a higher CBPscore had a poor prognosis which was verified by the Kaplan–Meier curve analysis.

According to calculated CBPscores, we classified these patients into CBP_low and CBP_high groups based on the median value, rather than using a prognosis-related optimal value, to enhance the randomness and accuracy. Patients in the CBP_low group had a longer OS and PFS in all training and validating cohorts. Furthermore, based on multivariate Cox regression analyses, the CBP signature was found as an independent risk factor for patients with GC. Furthermore, we proposed a nomogram by integrating three independent factors, including AJCC stage, age and CBP signature, and the calibration chart revealed that the nomogram had a better predictive value.

Considering the subcellular localizations that nine of the ten CBP signature genes located in extracellular space [14], we predicted a tight association between the CBP signature and tumor microenvironment. Then, an ESTIMATE method was used to analyze the immune scores and tumor purity. After that, we further analyzed the association between CBP signature and tumor immune scores. Interestingly, the CBP signature was found significantly correlated to immune scores and tumor purity. To confirm the association between the CBP signature and immune microenvironment, 22-type infiltrating immune cells in each tumor were analyzed using a CIBERSORT algorithm to further identify the immune features in different CBP groups. Then, the results revealed that there was a higher ratio of anti-tumor immune cells, which may be activated under immunotherapy, in the CBP_low group, while more pro-tumors cells in the CBP_high group. To verify this, we performed mIF in pathological resections from four responders and four non-responders following immunotherapy; the results found that responders had more anti-tumor immune cell infiltrations. These results confirmed the association between the CBP signature and tumor immune microenvironment.

Interestingly, the present research also highlighted the potential role of the CBP signature in predicting the response to immunotherapy. PD-1 and PD-L1 are co-stimulatory immune checkpoint targets and exert functions by activating anti-tumor immune cells, especially CD8+ T cells [36,37]. Then, we predicted that patients with more anti-tumor immune cells in the tumor nest had a stronger anti-tumor activity following immunotherapy. We first evaluated the relationship indirectly. We collected information on MSI, TMB and CBPscores. TMB and MSI-H are classic biomarkers to predict response for immunotherapy as neoantigen burden, which induces anti-tumor immune responses and is always increased by TMB and MSI-H [38,39,40,41]. Collectively, patients with high levels of TMB and MSI had a low CBP signature, which is useful for T cell recognition [42]. Then, this relationship was further verified in clinical data. The results revealed that responders to immunotherapy using PD-1 blockade had lower CBPscores. However, patients with lower CBPscores responded poorly to chemotherapy in our cohort. Therefore, the CBP signature is associated with tumor immune infiltration and can be used for guiding individual treatment for patients with GC.

Of course, there were several limitations in our study. First, cohorts in the present study were collected from different datasets; intratumor heterogeneity and interpatient heterogeneity were inevitable. Second, although survival impact and immune interaction of the CBP signature were found in the GC cohorts, the underlying molecular mechanisms behind these phenomena remain unclear. Additionally, the size of the patient group used for PD-1 block therapy was small and the predictive value of our CBP signatures needs further verification in larger prospective studies. Third, it is difficult to identify gastric cancer tissues undergoing cuproptosis using current techniques. Finally, applying the bioinformatic approaches in real-world applications remain challenging.

## 5. Conclusions

Herein, we performed a first CBP landscape analysis in patients with GC. A reliable, clinically feasible prognostic signature named CBP was established and the potential underlying immune-related mechanisms of this signature were identified. Importantly, a low CBPscore was tightly associated with well-validated immunotherapy biomarkers, TMB and MSI-H, and indicated a better response to PD-1 blockade. Thus, the CBP signature could be used as a tool for predicting prognosis and immunotherapy response in patients with GC. Future validation of the predictive capability of the CBP signature will be helpful for patients seeking counseling and individualized treatment.

## Figures and Tables

**Figure 1 biomolecules-12-01459-f001:**
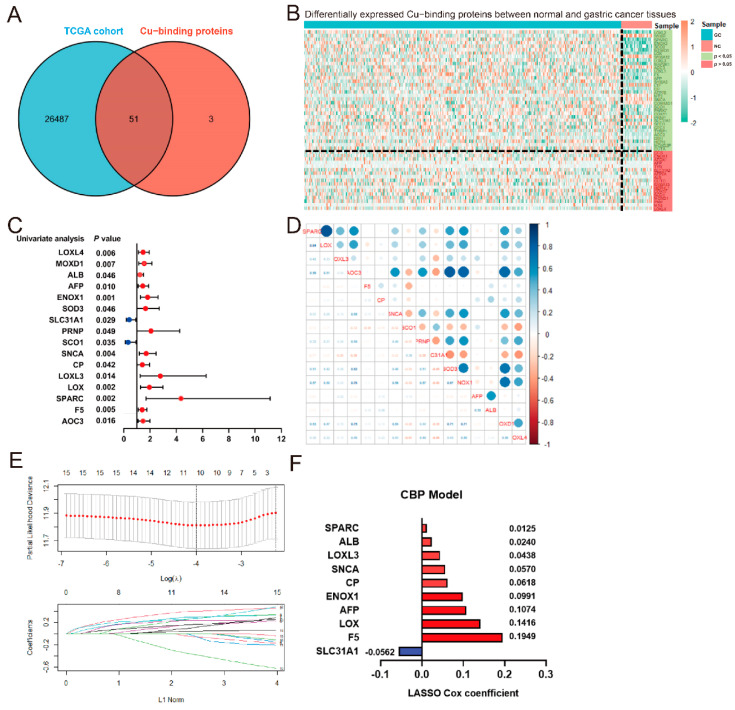
Construction of the Cu−binding proteins (CBP) signature. (**A**) The Venn plot of the human copper proteome and the mRNA expression profile from the TCGA cohort. (**B**) Differentially expressed CBP between GC and normal tissues from the TCGA cohort. (**C**) The univariate Cox analysis found 16 prognostic CBP in GC cases. (**D**) The linear correlation analysis among 16 prognostic CBP in GC. (**E**) The most robust predictive genes were identified using the least absolute shrinkage and selection operator Cox regression algorithm. (**F**) An ensemble of 10 genes remained with non-zero coefficients.

**Figure 2 biomolecules-12-01459-f002:**
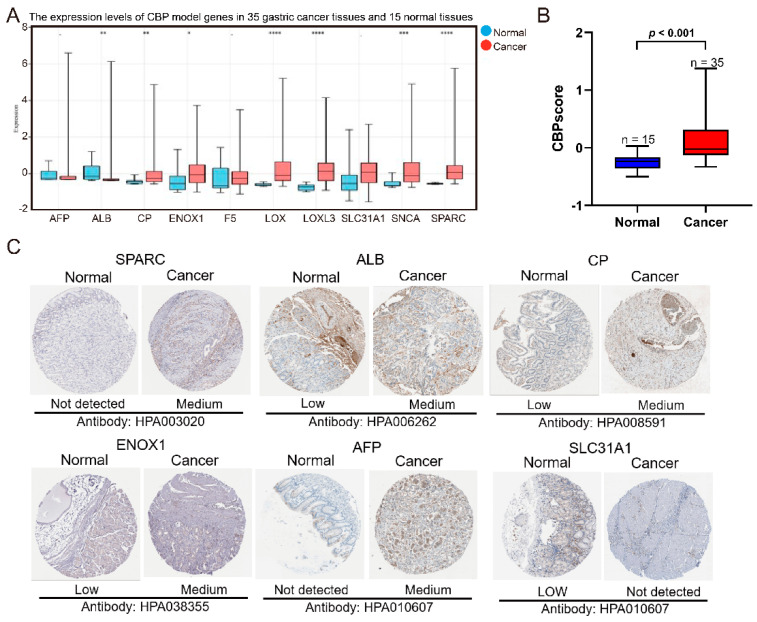
The verification of RNA and protein expression levels of the 10 markers of the CBP signature. (**A**) The mRNA expression levels of CBP signature genes between 35 GC tissues and 15 normal tissues from PKUCH. (**B**) The calculated CBPscores in GC and normal tissues. (**C**) The immunohistochemistry staining images of SPARC, ALB, CP, ENOX1, AFP and SLC31A1 from the HPA database. * *p* < 0.05, ** *p* < 0.01, *** *p* < 0.001, **** *p* < 0.0001.

**Figure 3 biomolecules-12-01459-f003:**
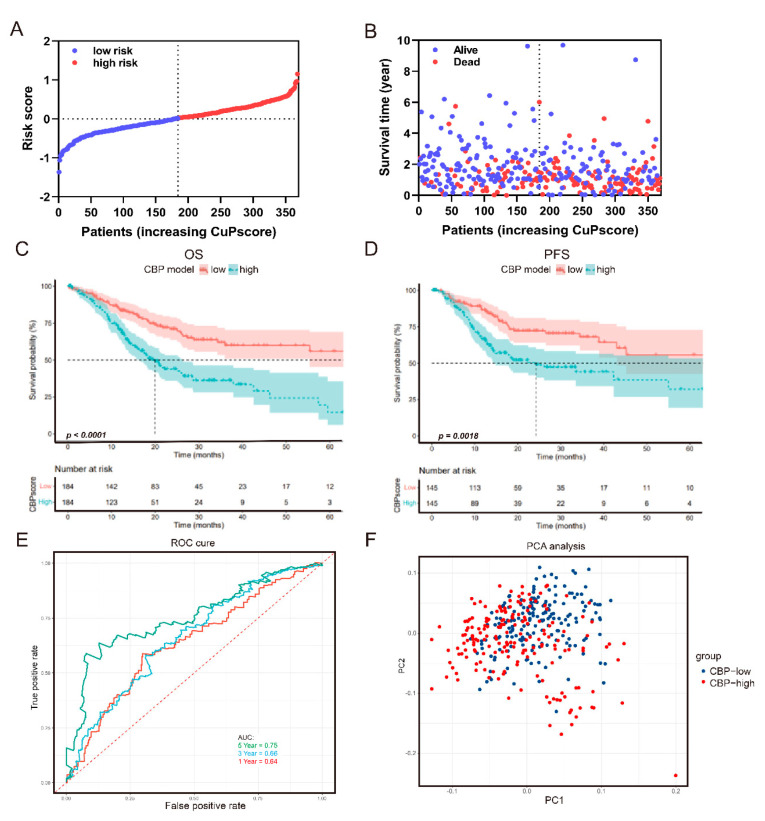
Prognostic analysis of the CBP signature in the TCGA set. (**A**,**B**) The distribution and median value of the CBP signature. (**C**,**D**) Kaplan−Meier curves for the OS and PFS of GC patients in the CBP_low group and CBP_high group. (**E**) AUC of time−dependent ROC curves at 1, 3 and 5 years. (**F**) PCA analysis showed different distribution patterns in the CBP_low group and CBP_high group.

**Figure 4 biomolecules-12-01459-f004:**
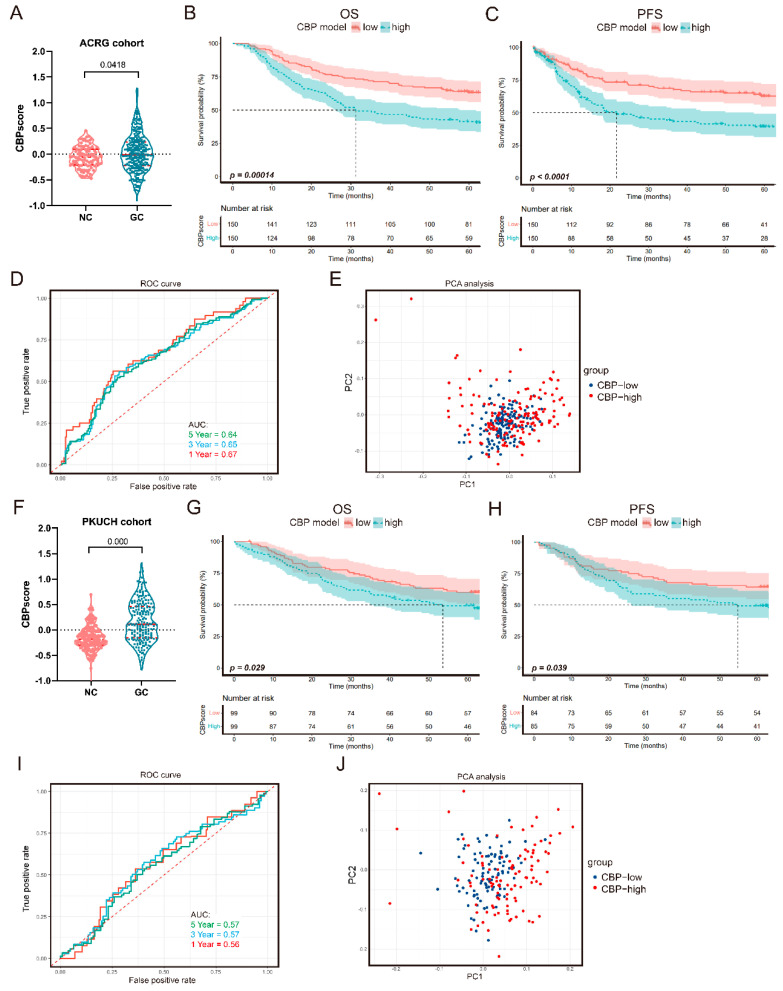
Validation of the CBP signature in the ACRG and PUKCH sets. (**A**) The scores of CBP signature in normal and GC tissues form the ACRG cohort. (**B**,**C**) Kaplan–Meier curves for the OS and PFS of GC patients in the CBP_low group and CBP_high group from the ACRG cohort. (**D**) AUC of time-dependent ROC curves at 1, 3 and 5 years in the ACRG cohort. (**E**) PCA analysis in the CBP_low group and CBP_high group from the ACRG cohort. (**F**) The scores of CBP signature in normal and GC tissues form the PKUCH cohort. (**G**,**H**) Kaplan–Meier curves for the OS and PFS of GC patients in the CBP_low group and CBP_high group from the PKUCH cohort. (**I**) AUC of time-dependent ROC curves at 1, 3 and 5 years in the PKUCH cohort. (**J**) PCA analysis in the CBP_low group and CBP_high group from the from the PKUCH cohort.

**Figure 5 biomolecules-12-01459-f005:**
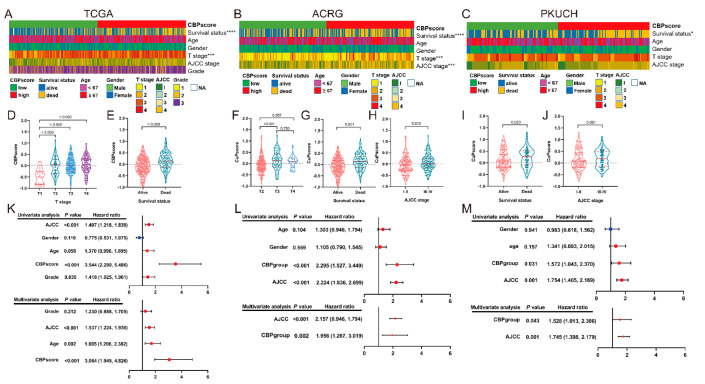
Relationships between the CBP signature and clinicopathological characteristics. (**A**–**C**) Heatmap of the clinicopathological characteristics and the scores of CBP signature; (**D**–**J**) the scores of CBP signature in different groups classified by clinical characteristics from the TCGA, ACRG and PKUCH sets. (**K**–**M**) Univariate and multivariate Cox regression analyses regarding OS in the TCGA, ACRG and PKUCH sets. * *p* < 0.05, *** *p* < 0.001, **** *p* < 0.0001.

**Figure 6 biomolecules-12-01459-f006:**
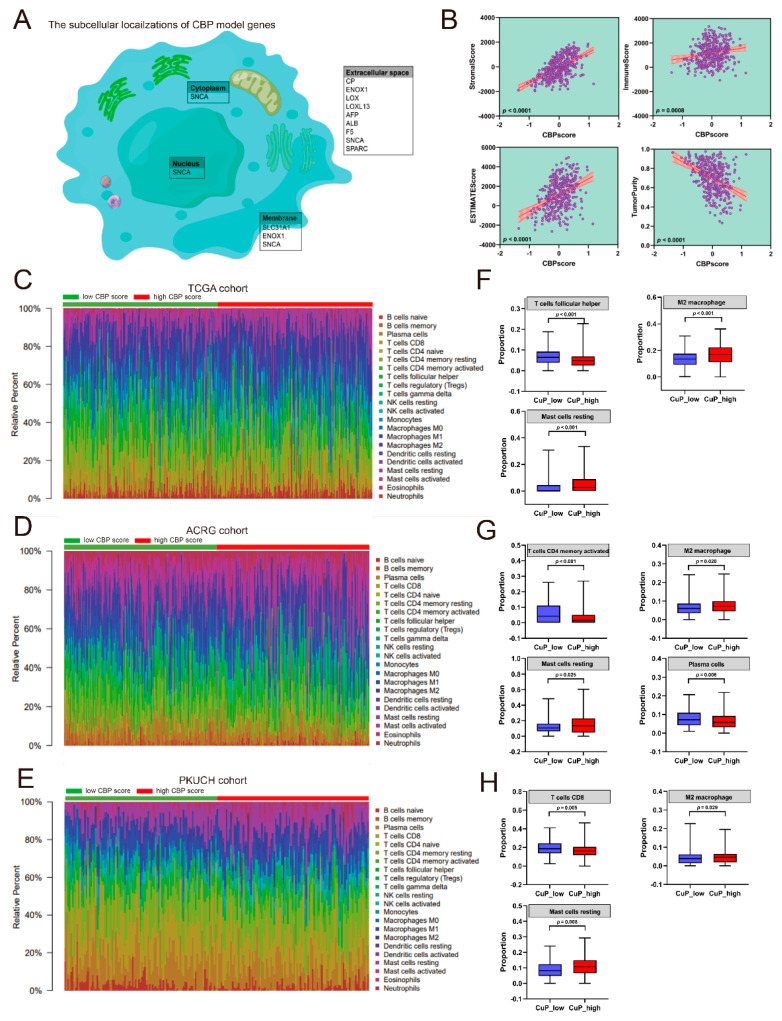
Immune infiltration analyses. (**A**) The subcellular localization of genes in the CBP signature. (**B**) Association between the CBP signature and stromal, immune scores and tumor purity. (**C**–**E**) Heatmap of the ratio of 22-type immune cells in GC patients in the low- or high-CBP subgroups from the TCGA, ACRG and PKUCH sets. (**F**–**H**) The different distribution of immune cells in the low- or high-CBP subgroups from the TCGA, ACRG and PKUCH sets.

**Figure 7 biomolecules-12-01459-f007:**
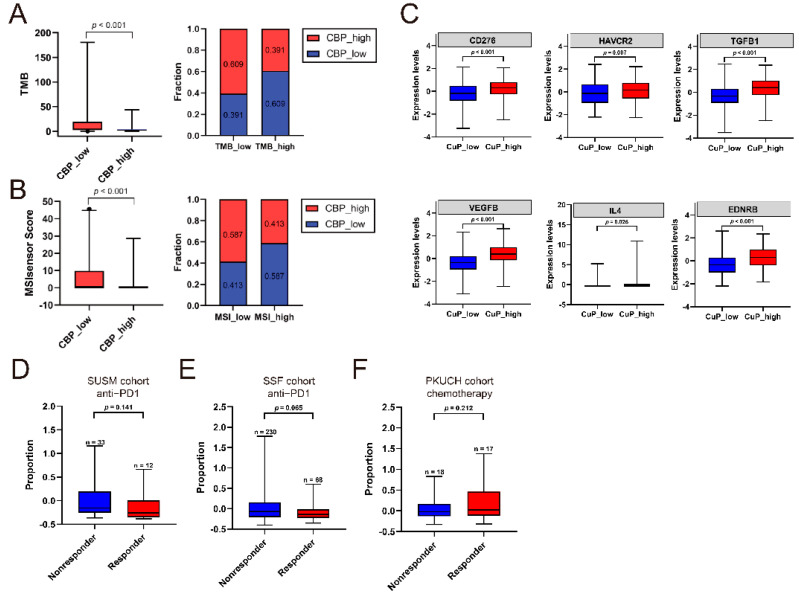
Relationships between the CBP signature, TMB, MSI and treatment response. (**A**) Association between the CBP signature and TMB in the TCGA cohort. (**B**) Association between the CBP signature and MSI in the TCGA cohort. (**C**) Association between the CBP signature and ACRG molecular subtypes in the ACRG cohort. (**D**) The scores of CBP signature in responders and non-responders to immunotherapy in the SUSM cohort. (**E**) The scores of CBP signature in responders and non-responders to immunotherapy in the SSF cohort. (**F**) The scores of CBP signature in responders and non-responders to chemotherapy in a PKUCH cohort.

**Figure 8 biomolecules-12-01459-f008:**
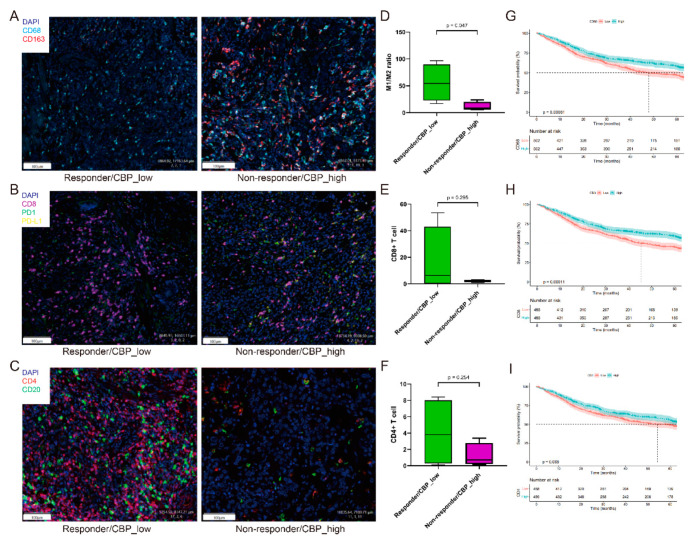
Immune infiltration analyses. (**A**–**C**) Detection of infiltrated macrophages, CD8+ T cells and CD4+ T cells in 4 responders/CBP_low and 4 non-responder/CBP_high to immunotherapy in the PKUCH cohort. (**D**–**F**) The M1/M2 ratio, CD8+ T cell proportion and CD4+ T cell proportion in 4 responders/CBP_low and 4 non-responders/CBP_high to immunotherapy. (**G**–**I**) Kaplan–Meier curves for the OS of GC patients in different groups classified by the median of CD68, CD8 and CD4 levels from a PKUCH cohort.

**Figure 9 biomolecules-12-01459-f009:**
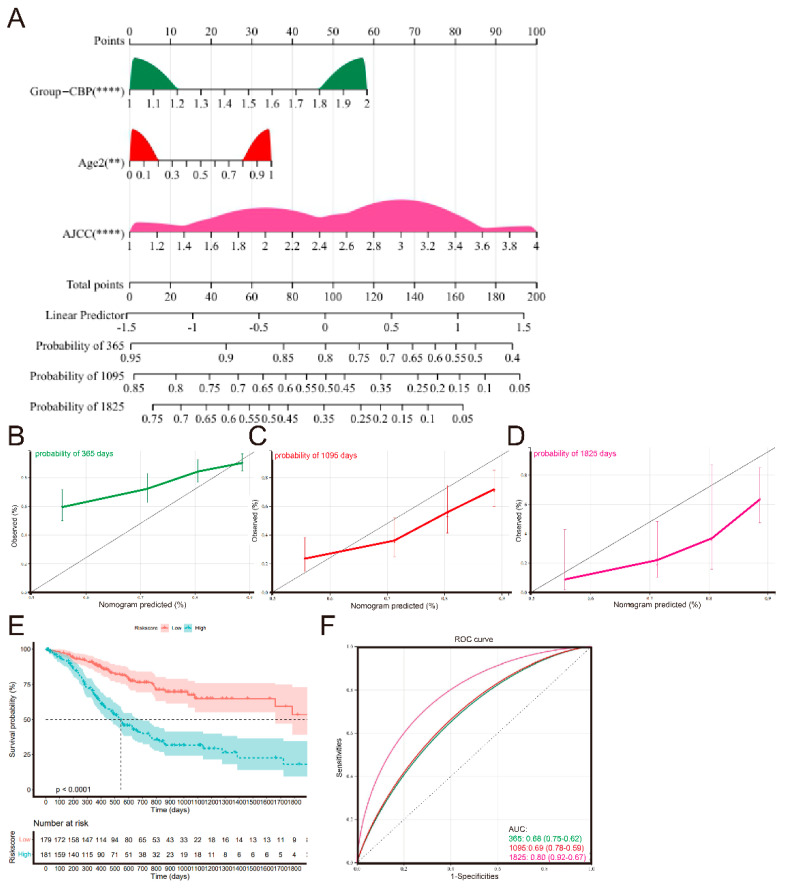
Nomogram to predict OS in GC patients. (**A**) Nomograms using prognostic factors identified by multivariate Cox analysis in the TCGA set. (**B**–**D**) The calibration curve for determining the reliability of the nomogram to predict the 1-year, 3-year and 5-year OS. (**E**) Kaplan–Meier curves for the OS of GC patients in the low nomogram score group and high nomogram score group in the TCGA cohort. (**F**) AUC of time-dependent ROC curves at 1, 3 and 5 years in the TCGA cohort. ** *p* < 0.01, **** *p* < 0.0001.

## Data Availability

The original data presented in this study are included in the article or Appendix A; other information can be directed to the corresponding authors.

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
