# Peer review of "Clinical Significance and Immune Infiltration Analyses of the Cuproptosis-Related Human Copper Proteome in Gastric Cancer"

_biomolecules, 2022, doi:10.3390/biom12101459_

Round 1

Reviewer 1 Report

You showed the importance of copper-binding protein as a prognostic factor in gastric cancer. Many data showed that the higher the CBP score, the worse the prognosis. You also found an association between CBP score and response to immunotherapy. I have several questions.

1.         Cuproptosis is a new cell death concept. Is it possible to detect cells that have undergone cuproptosis in gastric cancer tissue? Is it possible to distinguish cuproptosis from other cell deaths?

2.         Please describe the role of copper-binding proteins in cuproptosis. Do they promote or inhibit cuproptosis?

3.         Is cuproptosis more important than ferroptosis or pyroptosis in gastric cancer?

4.         Please describe the effect of free Cu on gastric cancer growth and cell death.

5.         Please discuss the mechanisms how free copper and copper-binding proteins affect cancer immunity.

6.         Please explain how you plan to apply these results to GC treatment.

Author Response

Reviewer 1:

You showed the importance of copper-binding protein as a prognostic factor in gastric cancer. Many data showed that the higher the CBP score, the worse the prognosis. You also found an association between CBP score and response to immunotherapy. I have several questions.

  1. Cuproptosis is a new cell death concept. Is it possible to detect cells that have undergone cuproptosis in gastric cancer tissue? Is it possible to distinguish cuproptosis from other cell deaths?

Reply: First, we would like to express our sincere thanks to you for the constructive and positive comments.

Cuproptosis is a new form of cell death triggered by targeted accumulation of Cu in mitochondria that drives lipoylated TCA cycle enzyme aggregation via direct Cu binding. We can detect the intracellular Cu ion concentration to understand the level of cuproptosis [1]. Cuproptosis can also be predicted by detecting several key molecules (FDX1, ATP7A, ATP7B, LIAS, DLD, SLC7A1 and LIPT1) and distinguished from other cell deaths [2]. In addition, measuring the activity TCA cycle also indirectly reacts to cuproptosis [2].

This study first analyzed the association between Cu-binding proteins and gastric cancer, and if the known cellular machinery and enzymatic targets for Cu can be used to identify gastric cancer tissue undergoing cuproptosis needs further verification. We are going to explore the specific method to identify cuproptosis in gastric cancer tissue in future studies. In addition, we have supplemented the related information in the discussion section.

  1. Please describe the role of copper-binding proteins in cuproptosis. Do they promote or inhibit cuproptosis?

Reply: Thanks very much for your excellent comments.

Cu is an essential element for most biological organisms, and acts as a cofactor for many enzymes. An appropriate amount of cellular free Cu is important for cell biology, but too much is toxic and can result in cuproptosis. To avoid the toxicity of free Cu, organisms have specific transport systems that ‘chaperone’ the metal to targets.

The human Cu proteome, Cu-binding proteins, include two main kinds of proteins: One is Cu-transporting proteins that participate in regulating Cu homeostases, such as CTR1 and CTR2 responsive for importing Cu, SLC31A1, CCS and ATOX1 responsive for transporting Cu to cytoplasmic, and COX11, COX17, SCO1, and SCO2 responsive for providing Cu to the mitochondria. Although there are several Cu-binding proteins responsive for exporting Cu, such as ATP7A and ATP7B, most proteins of the human Cu proteome increase cellular free Cu to promote cuproptosis. The other kind is Cu-dependent enzymes that serve as the downstream biological effects of Cu, which account for nearly half of the proteins identified to be able to bind with Cu. In addition, the Cu proteome also includes some Cu-binding proteins with unknown functions.

We have supplemented the information in the introduction section.

  1. Is cuproptosis more important than ferroptosis or pyroptosis in gastric cancer?

Reply: Thank you for your excellent comments.

Previous studies have revealed that ferroptosis and pyroptosis are important factors that impact the oncogenesis and progression of gastric cancer. Ferroptosis is an iron-dependent form of cell death. It is characterized by intracellular lipid peroxide accumulation and redox imbalance. Iron oxidation contributes to tumor formation and the development of cancer. Ferroptosis has also been found to be closely related to the proliferation, invasion, and metastasis of gastric cancer and understanding the processes underlying ferroptosis is promising for the development of cancer treatment strategies. Pyroptosis, a type of inflammatory programmed cell death, is mediated by multiple inflammasomes which can recognize danger signals and activate the secretion of pro-inflammatory cytokines. It can induce cancer cell death within the gastrointestinal tract. 

Cuproptosis is a new form of cell death identified in 2022 and the association between cuproptosis and gastric cancer was firstly analyzed in this study. We found that 31 (62.0%) Cu-binding proteins were dysregulated in gastric cancer and 16 of them are associated with the survival of gastric cancer patients. The importance of cuproptosis, ferroptosis and pyroptosis in gastric cancer need further exploration.

  1. Please describe the effect of free Cu on gastric cancer growth and cell death.

Reply: Copper (Cu) is essential for living organisms and acts as a cofactor in many metabolic enzymes, but too much is toxic and can result in cuproptosis. To avoid the toxicity of free Cu, organisms have specific transport systems that ‘chaperone’ the metal to targets. Previous studies revealed that the overload of free Cu promotes cell death and inhibits gastric cancer growth [3, 4]. Besides, a previous study found that nontoxic concentrations of Cu also enhanced the inhibitory effects of disulfiram on gastric cancer cell viability and colony formation [5].

Collectively, increased intracellular free Cu exerts cytotoxic effects and inhibits gastric cancer growth.

  1. Please discuss the mechanisms how free copper and copper-binding proteins affect cancer immunity.

Reply: Thanks very much for your excellent comments.

As a cofactor for many metabolic enzymes, intracellular free copper (Cu) plays an important role in regulating cell state, including cancer cells. The recent finding revealed that Cu plays an important role in regulating the tumor immune microenvironment. Intratumor Cu can modulate PD-L1 expression to influence tumor immune evasion [6]. Besides, the Cu-mediated biological process influences immunogenic cell death and immune cell infiltration of tumors [7]. In the current study, the CBP (Cu-binding proteins, the human Cu proteome) signature was found to be associated with infiltrated intratumor immune components. We found that GC patients with high CBP signature scores exhibited increased infiltration of protumor immune components, whereas reduced fraction of antitumor immune cells such as CD8+ T cells, CD4+ T cells and B cells. Thus, it is highly valuable to explore the underlying molecular mechanisms between Cu levels and tumor immune infiltration.

  1. Please explain how you plan to apply these results to GC treatment?

Reply: Thanks very much for your excellent comments. The findings of cuproptosis-related Cu-binding proteins in this study have several main applications for GC treatment. First, the established CBP signature can predict the prognosis of patients with GC. Second, it can also be used for guiding individual treatment for patients with GC, as patients with lower CBPscores respond poorly to chemotherapy but respond well to immunotherapy. Third, these CBP proteins may be underlying targets to block or eradicate GC. In addition, combining Cu-related treatment may be more effective in increasing the efficacy of immunotherapy for GC patients.

Finally, we sincerely thank you for your excellent comments again! These comments not only let us have a deeper understanding of the roles of copper and cuproptosis in GC, but also point to a direction for our further research.

References

[1] Oka, et al. An immunohistochemical study of copper, zinc-containing superoxide dismutase detected by a monoclonal antibody gastric mucosa and gastric cancer. Histopathology. 1990. 17(3): 231-236.

[2] Tsvetkov, et al. Copper induces cell death by targeting lipoylated TCA cycle proteins. Science. 2022, 375(6586): 1254-1261.

[3] Xia, et al. A new Schiff base coordinated to copper (II) compound induces apoptosis and inhibits tumor growth in gastric cancer. Cancer Cell Int, 2019. 19: 81.

[4] Badrooh, et al. Trigger of apoptosis in adenocarcinoma gastric cell line (AGS) by a complex of thiosemicarbazone and copper nanoparticles. Mol Biol Rep, 2022. 49(3): 2217-2226.

[5] Cheng Du, et al. Disulfiram/copper induces antitumor activity against gastric cancer cells in vitro and in vivo by inhibiting S6K1 and c-Myc. Cancer Chemother Pharmacol. 2022 Apr;89(4):451-458.

[6] Voli, et al. Intratumoral Copper Modulates PD-L1 Expression and Influences Tumor Immune Evasion. Cancer Res. 2020. 80(19): 4129-4144.

[7] Zheng, et al. Remodeling tumor immune microenvironment (TIME) for glioma therapy using multi-targeting liposomal codelivery. J Immunother Cancer. 2020. 8(2).

Reviewer 2 Report

Tang et el used bioinformatic approaches and machine learning algorithm to identify CBP signature for predicting treatment response in patients with gastric cancer (GC). While the authors have used 946b GC patient data to identify CBP signature genes as a factor in disease prognosis, some of the findings (in particular, response to PD-1 block therapy) were validated with the response from 8 patients only (4 responders and 4 non-responders). While the data looks promising for a small group of patients and the overall study design is sound for such studies, the authors should include a limitation statement in the conclusion/discussion section citing the small size of the patient group used for PD-1 block therapy and the limitations of bioinformatic approaches in real world applications.

Minor corrections:

-Line 16: Remove “s” from bioinformatics.

-Line 16: Replace “was” with “were”.

-Figure labels are not legible in some figures (e.g.,1D, E, 2A, 3, 4B-E, G-J, 8, 9)  whereas they are legible in some others suggesting that these figures were generated by different individuals. Most of the figures, in their current forms, are not of publication quality. The authors should fix these figures before the manuscript can be published.

-Supplementary image quality is very poor…labels are not legible. Replace this figure with a higher resolution one.

References 

-First letter of each word is capitalized in the title of a number of references whereas lower case letters were used in others…be consistent in using upper/lower case letters.

- Reference 42: “Cancer Immunology” is not a part of the title of the article cited. Remove it.

Author Response

Reviewer 2:

Comments and Suggestions for Authors

Tang et el used bioinformatic approaches and machine learning algorithm to identify CBP signature for predicting treatment response in patients with gastric cancer (GC). While the authors have used 946b GC patient data to identify CBP signature genes as a factor in disease prognosis, some of the findings (in particular, response to PD-1 block therapy) were validated with the response from 8 patients only (4 responders and 4 non-responders). While the data looks promising for a small group of patients and the overall study design is sound for such studies, the authors should include a limitation statement in the conclusion/discussion section citing the small size of the patient group used for PD-1 block therapy and the limitations of bioinformatic approaches in real-world applications.

Reply: First, I want to thank you for the extremely helpful comments provided for our paper. In the paper, we’ve addressed all comments.

Supplementing the limitation statement in the discussion section is very meaningful for the current manuscript. We have rechecked this study and summarized several limitations in the revised manuscript. The detailed informations are as follows:

“Of course, there were several limitations in our study. First, cohorts in the present study were collected from different datasets, intratumor heterogeneity and interpatient heterogeneity were inevitable. Second, although survival impact and immune interaction of the CBP signature were found in the GC cohorts, the underlying molecular mechanisms behind these phenomena remained unclear. Besides, the size of the patient group used for PD-1 block therapy was small and the predictive value of our CBP signatures needs further verification in larger prospective studies. Third, it is difficult to identify gastric cancer tissues undergoing cuproptosis using current techniques. Finally, applying the bioinformatic approaches in real-world applications remain challenging.”

Minor corrections:

-Line 16: Remove “s” from bioinformatics.

-Line 16: Replace “was” with “were”.

Reply: Thank you for your excellent comments and we have corrected them.

-Figure labels are not legible in some figures (e.g.,1D, E, 2A, 3, 4B-E, G-J, 8, 9), whereas they are legible in some others suggesting that these figures were generated by different individuals. Most of the figures, in their current forms, are not of publication quality. The authors should fix these figures before the manuscript can be published.

Reply: Thank you for your excellent comments. We have reedited these figure labels by one author.

-Supplementary image quality is very poor…labels are not legible. Replace this figure with a higher resolution one.

Reply: Thank you for your excellent comments. We have resubmitted the supplementary image with 1200 dpi and the label has been rewritten.

References 

-The first letter of each word is capitalized in the title of a number of references whereas lower case letters were used in others…be consistent in using upper/lower case letters.

- Reference 42: “Cancer Immunology” is not a part of the title of the article cited. Remove it.

Reply: Thank you for your excellent comments. We have rechecked the references and made them consistent in using upper/lower case letters. In addition, we have removed “Cancer Immunology” in Reference 42.

Finally, we sincerely thank you for your excellent comments again! These comments not only let us have a deeper understanding of the roles of copper and cuproptosis in GC, but also point to a direction for our further research.

Round 2

Reviewer 1 Report

The manuscript was improved.